# Mapping of sex work hotspots to guide targeted HIV prevention: Evidence from eight Ukrainian cities

Oksana Kovtun[1]*, Olga Cheshun[1], Oksana Pashchuk[1], Kostyantyn Dumchev[2]

1 Alliance for Public Health, Kyiv, Ukraine, 2 National University of Kyiv-Mohyla Academy, Kyiv, Ukraine

* kovtun@aph.org.ua

## Abstract

### Background

Ukraine's HIV epidemic remains concentrated among key populations, with sex workers (SWs) facing overlapping vulnerabilities, including a history of injection drug use. Although overall HIV prevalence has declined, prevention coverage remains below national and global targets. Accurate mapping of sex work hotspots is essential for effective outreach, yet existing data are fragmented and incomplete, especially in privatized and digitally mediated environments. This study aimed to systematically identify and characterize sex work venues across eight Ukrainian cities and assess their coverage by HIV prevention services.

### Methods

In 2021, we conducted a two-stage, multi-informant mapping study across eight Ukrainian cities, interviewing 1,212 secondary and 2,277 primary key informants, including SWs. Hotspots were characterized by type, perceived safety, access modality, operating schedule, and presence of SW subgroups. We used descriptive statistics, exploratory factor analysis, and multivariable mixed-effects regression to identify factors associated with HIV prevention service coverage.

### Results

Of 2,581 identified hotspots, 2,118 (82.1%) were validated as active. Apartments (43.2%), virtual platforms (11.7%), and street-based sites (11.1%) were the most common. Only 13.7% of hotspots were registered in the national HIV service registry, while 34.1% received prevention service delivery and 18.9% were reached by mobile vans. Coverage was highest at street-based and office-type venues and consistently lowest in private, virtual, and pimp-mediated settings. Service coverage was positively associated with 24/7 or daytime operation, higher perceived safety, open access, and the presence of SWs who inject drugs.

**Data availability statement:** The study did not use third-party data. All data were collected and are owned by the International Charitable Foundation "Alliance for Public Health" (APH). Due to the sensitive nature of the data, which include information on key populations and specific sex work locations, the datasets cannot be made publicly available in order to protect participant confidentiality and safety, in accordance with APH data protection policies and ethical requirements. De-identified data may be made available upon reasonable request and subject to institutional approval. Requests for data access should be directed to the Alliance for Public Health at office@aph.org.ua.

**Funding:** This study was conducted with the financial support of the International Charitable Foundation "Alliance for Public Health" through the project "Improving HIV Treatment Cascade for Key Populations through Differentiated Case Detection and Linkage to Care and Increased Capacity at the Center for Public Health and Strategic Information in Ukraine", funded by the U.S. Centers for Disease Control and Prevention (CDC) through the US President's Emergency Plan for AIDS Relief (PEPFAR). This study is supported by the Cooperation Agreement NU2GGH002114 with the CDC. All responsibility for the content shall be borne by the authors, and the article does not necessarily reflect the official position of the CDC.

**Competing interests:** The authors have declared that no competing interests exist.

## Conclusions

This study provides comprehensive mapping of sex work hotspots in Ukraine, revealing persistent gaps in HIV prevention for less visible and harder-to-reach SWs. Community-led mapping achieved high validation rates (86.8%) and identified substantial unmet needs in digital and concealed hotspots. Routine hotspot mapping, combined with engagement with SW communities, is essential for data-driven, equitable, and adaptive HIV responses in rapidly changing contexts.

## Introduction

Despite ongoing HIV prevention and care efforts [1], Ukraine's epidemic remains concentrated, with sex workers (SWs) among the most affected key populations [2]. Although the early epidemic was driven primarily by injection drug use, heterosexual transmission has since become dominant [3], accounting for 64.9% of new HIV diagnoses in 2022 [4]. This shift is partly driven by overlapping vulnerabilities between people who inject drugs (PWID) and SWs [1,5], including SWs with injection histories and clients who use drugs [6]. As a result, SWs continue to represent a priority population for HIV prevention and testing programs.

In 2021, an estimated 78,385 SWs were active in Ukraine [7]. Although overall HIV prevalence among SWs declined from 5.2% in 2017 [8] to 3.1% in 2021, prevalence remained high (24.1%) among SWs with a history of injection drug use [9]. This decline likely reflects a combination of mortality, transitions out of sex work, and the expansion of community-based prevention programs led by non-governmental organizations (NGOs) [10]. Nevertheless, HIV prevention coverage among SWs remains below both national and global targets. Program reach increased from approximately 38,000–55,000 SWs between 2017 and 2021 [4], yet the National HIV Strategy aims for 70% coverage [11], while UNAIDS sets a target of 95% by 2025 [12]. Although HIV treatment cascade indicators among SWs improved – from 47% diagnosed, 59% on antiretroviral therapy, and 86% virally suppressed in 2017 [8] to 83-94-80 in 2021 [9] – they still fall short of the 95-95-95 goals. These gaps highlight the need to strengthen HIV testing, which remains a critical entry point to both prevention and treatment services [13], particularly in the context of ongoing war-related population displacement [14] and increased mobility of SWs [15,16], which have disrupted established outreach routes and complicated sustained service delivery across settings.

Delivering HIV prevention and testing services at sex work hotspots through outreach teams or mobile health clinics (vans) is an evidence-based strategy used across diverse settings [17–22]. In Ukraine, the first mobile van was launched in 2004, and rapid HIV testing was introduced in 2006 [23]. These approaches help overcome barriers commonly associated with facility-based services, such as long travel distances, limited awareness of service locations, long waiting times, and inconvenient operating hours [17,24,25]. However, their effectiveness depends on having accurate, up-to-date information about where SWs are actually located.

Hotspot mapping is therefore a foundational component of outreach- and mobile van-based HIV programming [26–29]. It is also essential for time-location sampling frames in integrated bio-behavioral surveys (IBBS), population size estimation (PSE) [30,31], and the design of geographically and temporally appropriate interventions [32–34]. Location-specific and venue-sensitive analyses are also needed to guide effective outreach planning [35], particularly for less visible, informal, and digitally mediated settings [36,37], which is important in the context of uneven and low HIV prevention coverage among SWs [38]. In Ukraine, mapping is typically conducted during IBBS preparation but is rarely validated (often <20%) or used beyond sampling frame development [8,39–44]. Moreover, hotspot data are seldom systematically analyzed or applied to guide service delivery [45]. While individual-level HIV prevention coverage among SWs has been routinely reported through programmatic and surveillance data [46], evidence on prevention service coverage at the level of sex work hotspots in Ukraine remains largely absent, particularly in relation to venue characteristics and service delivery contexts.

To address this gap, we conducted a comprehensive hotspot mapping exercise across eight Ukrainian cities as part of the preparation for the 2021 IBBS round. While the primary purpose of the mapping was to inform recruitment and guide PSE, the resulting dataset also provided an opportunity for systematic analysis to support practical decision-making in HIV prevention – an approach that remains uncommon in Ukraine and is rarely described in the published literature as a basis for routine HIV prevention programming. Beyond identifying and validating active sex work venues, the mapping captured detailed information on hotspot typologies, operating schedules, access modalities, perceived safety, subgroup presence, and outreach activity. This enabled not only a descriptive characterization of where and how sex work is organized, but also an examination of how these operational features relate to HIV prevention service delivery at the hotspot level. By generating granular, context-sensitive data, we aimed to support geographic prioritization, strengthen outreach planning, and improve the equity and efficiency of HIV services for SWs – particularly in the context of limited resources and a dynamic epidemic landscape. Accordingly, this analysis was guided by two research questions: (1) What operational typologies of sex work hotspots can be identified in urban Ukraine based on patterns of sex work organization? (2) Which hotspot characteristics are associated with HIV prevention service coverage?

## Materials and methods

### Population and settings

Hotspot mapping was conducted between March and July 2021 in eight Ukrainian cities (Cherkasy, Dnipro, Kharkiv, Kropyvnytskyi, Kyiv, Lviv, Mariupol, and Odesa), selected based on HIV prevalence among SWs in 2008–2017, in consultation with the National IBBS Working Group led by the Public Health Center of the Ministry of Health. Table 1 presents city-specific HIV and HCV prevalence and 2019 PSE among SWs.

SWs were defined as adults who exchanged sex for money, goods, or services. Individuals under 18 engaged in commercial sex were classified as sexually exploited minors (SEMs), in line with UNAIDS [47] and WHO [13] guidelines.

### Mapping methodology

We adapted an established approach [34,49,50] to include less visible forms of sex work, such as escort services, virtual platforms, and pimp-managed networks. Level 1 (L1; March 17 – April 14, 2021) focused on hotspot identification; Level 2 (L2; June 21 – July 31, 2021) validated active hotspots. A two-month delay between phases was due to COVID-19 restrictions in Ukraine [51].

**Hotspot definitions and classification.** A hotspot was defined as any physical venue or communication means where SWs meet or arrange to meet clients. Thirteen types were identified: apartments, massage parlors/saunas, cafes/bars, hotels/motels, offices ('brothels'), entertainment hotspots (nightclubs, casinos), art clubs/strip clubs, streets/parks, routes/highways, virtual hotspots (websites, apps, social media), escort/call services, pimp-mediated venues, and printed advertisements (flyers, posters, newspapers).

**Table 1. HIV prevalence among SWs, HCV prevalence, and population size estimates.**

| City | HIV Prevalence | | | | | SWs-PWID [8] | SWs-not PWID [8] | HCV Prevalence among SWs [8] | PSE, 2019 [48] |
|---|---|---|---|---|---|---|---|---|---|
| | Among SWs | | | | | | | | |
| | 2008 [43] | 2011 [42] | 2013 [41] | 2015 [44] | 2017 [8] | | | | |
| | % | % | % | % | % | % | % | % | N |
| Cherkasy | 17.9 | 14.4 | 0.0 | 18.6 | 26.7 | 46.0 | 36.6 | 36.6 | 1,300 |
| Dnipro | 14.0 | 9.6 | 6.7 | 2.2 | 2.6 | 11.2 | 0.9 | 6.4 | 4,000 |
| Kharkiv | 0.0 | 0.0 | 1.2 | 0.9 | 0.0 | 0.0 | 0.0 | 0.0 | 4,500 |
| Kropyvnytskyi | 17.0 | 13.7 | 6.0 | 6.5 | 10.9 | 17.0 | 6.8 | 39.2 | 1,400 |
| Kyiv | 24.7 | 24.2 | 2.0 | 1.9 | 6.9 | 48.1 | 1.9 | 15.8 | 14,600 |
| Lviv | 9.0 | 5.7 | 2.5 | 1.4 | 0.0 | 32.0 | 0.0 | 4.2 | 2,600 |
| Mariupol | 14.0 | 9.6 | 6.7 | 2.2 | 18.3 | 41.0 | 10.6 | 24.4 | 3,000 |
| Odesa | 16.5 | 13.5 | 10.3 | 9.5 | 2.9 | 9.0 | 2.5 | 1.8 | 10,100 |

SW, sex worker; PWID, people who inject drugs; PSE, population size estimate.

Each type definition was based on observable features and operating characteristics. For instance, 'offices' referred to brothel-like venues with multiple rooms, bar areas, and more than three SWs under managerial supervision; managers (often supported by administrators or security staff) coordinated client flow, and owners retained part of SW revenue [39]. 'Apartments' were private residences where one or more SWs worked independently, with or without living on-site. Virtual hotspots had no fixed location and involved online or phone-based arrangements.

## Level 1: Hotspot identification

Before fieldwork, 56 in-depth interviews with SWs, NGO outreach workers, healthcare providers, and HIV service staff were conducted to refine typologies, tools, and secondary key informants (KIs) list. Secondary KIs were purposively selected based on their routine contact with sex work settings or SWs in each city, with the aim of capturing both visible and less visible forms of sex work. These included SWs themselves, NGO and outreach staff, healthcare providers, venue staff (e.g., administrators or security personnel), taxi and truck drivers, law enforcement officers, and other individuals with contextual knowledge of sex work locations.

The list of secondary KIs was developed in collaboration with local NGOs and SW community representatives and was expanded iteratively through referrals during fieldwork. Hotspots were identified using a combination of HIV prevention program data and interviews with secondary KIs across 48 administrative districts. A list of hotspots where HIV services were delivered in 2020 was obtained from SYREX [52], the national HIV services monitoring system [53]. Additionally, secondary KIs were asked to name hotspots, and if agreed, to provide hotspot names, addresses or GPS coordinates, operational hours, and SW estimates until saturation (i.e., when no new hotspots were identified in subsequent interviews).

Virtual hotspots were identified through keyword searches on digital platforms. All data were stored in a centralized database and checked for duplicates. Data collection instruments are available in S1 File.

## Level 2: Hotspot validation

In L2, field teams visited each L1-identified hotspot to confirm SW presence and collect site-specific data. The teams first attempted to interview SWs (primary KIs); if unavailable, interviews were conducted with knowledgeable staff (e.g., security or administrators). For hard-to-reach venues (e.g., escort or virtual), off-site interviews were arranged. Access to unsafe sites was supported by peer workers, NGOs, or venue staff. Visits lasted 30 minutes and included SW headcounts

and hotspot solicitation until saturation. Newly named locations were validated through the same process. Hotspots with no confirmed SW presence after three different visits were classified as non-functional and excluded from analysis.

## Study team

The National IBBS Working Group oversaw all stages of the study. In each city, regional technical working groups coordinated implementation and included public health officials, NGO staff, researchers, and SW community members. SWs were involved in all stages – from planning and data collection to interpretation – in line with community engagement standards [30,40].

Each regional group comprised approximately 8–14 members, with 2–4 SW representatives, forming field teams together with local NGO staff and researchers. Across all cities, 15 NGOs and 81 trained data collectors participated. A training-of-trainers model was used: national trainers prepared regional coordinators and data managers, who then trained city-level teams, supported by the Alliance for Public Health.

## Data management and quality control

Field teams collected data using standardized paper forms and entered them daily into a password-protected Excel database. Quality control included review for missing values, logical inconsistencies (e.g., venues open more than 24 hours), and transcription errors. A 20% random sample per city was cross-verified against original forms. Duplicates were identified using normalized addresses (e.g., street name, building number) and GPS coordinates (within 50 meters) and reviewed by regional coordinators and SW representatives.

While no national monitoring visits occurred, regional coordinators and technical group members ensured protocol adherence and submitted weekly progress reports to the national team.

## Data analysis

Only L2-validated hotspots were included in the analysis. Characteristics of interest included hotspot type, working schedule, access modality (open vs. gatekeeper), perceived safety (Likert scale), number of SWs, and subgroup presence (PWID, male SWs, SEMs). Coverage of HIV prevention services was assessed using both registration in the national SYREX registry and self-reported service delivery. SW population estimates per hotspot were calculated as the midpoint between minimum and maximum estimates; direct headcounts were also recorded. City-level estimates were the sum of midpoints. As this study represents a full census of the target population, descriptive statistics are reported without confidence intervals or p-values, as the observed values represent the actual population parameters rather than sample estimates.

Hotspot working schedule was coded as a binary vector across 168 hourly time slots (from Monday 00:00–01:00 to Sunday 23:00–24:00). Based on operating hours during 06:00–18:00 (DayHours) and 18:00–06:00 (NightHours), hotspots were categorized as 'predominantly daytime', 'predominantly nighttime', or 'balanced'. Hotspots operating continuously were classified as '24/7'. To assess outreach potential, a heatmap was generated by calculating the proportion of open venues per hour and hotspot type, and categorizing hours into qualities (very low to high).

Because sex work in urban Ukraine is often organized across more than one venue type, single-category hotspot classification may not fully capture the operational contexts encountered by outreach teams. To address this gap and summarize overlapping work patterns in a way that is meaningful for service planning, we applied Exploratory Factor Analysis (EFA) [54] to cross-type hotspot data. To operationalize research question 1, hotspots were categorized as exclusive-type (SWs work at one venue type) or cross-type (SWs reported to work across multiple types). Cross-type status was derived from aggregated reports and does not reflect individual mobility. We then conducted EFA using principal axis factoring with Varimax rotation on 12 binary indicators of cross-type working to identify latent patterns in how sex work is organized across different hotspot types. Factors with eigenvalues >1 were retained, and factor loadings ≥0.40 were considered salient.

To address research question 2, we fitted a multivariable Generalized Linear Mixed Model [55] with a binomial distribution and logit link to test which hotspot characteristics were associated with the likelihood of HIV prevention service coverage. The dependent variable was self-reported HIV prevention service coverage at the hotspot level (yes/no), while independent variables included hotspot types (as defined by EFA), and characteristics hypothesized to influence service access, such as 24/7 availability, schedule pattern, number of SWs present during validation, presence of SEMs, perceived safety, access modality, and presence of PWID or male SWs. To account for within-city correlation, city was modeled as a random effect.

Data cleaning and analysis were performed using Microsoft Excel and IBM SPSS Statistics 28.0 (IBM SPSS Statistics for Windows, Version 28.0. Armonk, NY: IBM Corp).

## Ethical considerations

Ethical approval was obtained from the Institutional Review Board of the Alliance for Public Health (Kyiv, Ukraine, IRB#00012333, FWA#00027488) and reviewed by the CDC's human research protection office. CDC staff approved the protocol but did not participate in fieldwork or contact with respondents.

All national team members completed Collaborative Institutional Training Initiative (CITI) certification. Regional field staff completed the National Institutes of Health course and were trained in study protocols, data security, and participant safety.

Given the criminalization of sex work in Ukraine [56], field teams carried official IDs, and local law enforcement was notified in advance. Each hotspot received a unique anonymized code, and no personal identifiers were collected. Verbal informed consent was obtained from all participants, and participation was voluntary.

All KIs and SWs were assigned non-traceable codes. Paper forms were stored in locked cabinets, and digital records were encrypted and stored on password-protected servers with restricted access.

## Results

During L1, 1,212 secondary KIs identified 3,517 potential hotspots in total; 45.6% were SWs and 18.2% were taxi/truck drivers (Table 2). At L2, 2,581 primary KIs were interviewed on-site, 80.6% of whom were SWs. Validation rates across KI types ranged from 62.4% to 86.8%. Hotspot reports reflected KIs' professional context: e.g., hotel staff primarily listed hotel sites, while AIDS center staff most often named street-based venues. Apartments were the most frequently identified type.

Among the 3,517 identified hotspots, 2,581 were unique. Of them, 2,118 (82.1%) were confirmed active (Table 3). The only printed advertisement hotspot failed validation. Apartments comprised 43.2% of hotspots, followed by virtual platforms (11.7%) and street-based sites (11.1%). City-level distributions are provided in S1 Table. Most venues (94.4%) operated year-round, though winter street activity dropped to 71.2%. Escort/on-call venues had the highest 24/7 availability (94.8%), while cafés/bars had the lowest (7.4%). Overall, 63.0% had balanced schedules, 9.2% were daytime, and 27.8% predominantly nighttime venues. A heatmap, provided in S2 Table, showed extended operating hours for office ('brothel'), virtual, and hotel/motel hotspots.

Open access was reported at 88.2% of hotspots; however, entry and contact were often mediated informally by SWs or trusted intermediaries, even in venues classified as "open". Unsafe conditions (6.8%) – based on observed and/or reported violence or substance use – were concentrated at route/highway hotspots (34.4%) and cafés/bars (21.5%). SWs who inject drugs were reported at 8.3% of hotspots – mostly outdoors and in entertainment venues – and male SWs were present at 4.6%, primarily in pimp-mediated settings.

Hotspots were classified as either exclusive-type (52.6%) or cross-type (47.4%), with cross-venue work most common in pimp-mediated and escort/on-call settings. EFA produced five factors (Kaiser-Meyer-Olkin measure of sampling adequacy = 0.540; Bartlett's test of sphericity $\chi^2 = 1985.2$, df = 66, p < .001), which represent latent groupings based on

**Table 2. Typology of key informants and number of identified hotspots across eight Ukrainian cities, 2021.**

| KI Type | L1 | | | L2 | |
|---|---|---|---|---|---|
| | Secondary KIs | Reported Hotspots | Most Frequently Reported Type | Active Hotspots | Primary KIs |
| | n (column %) | n (column %) | – | n (% validated) | n (column %) |
| Total (N) | 1,212 | 3,517 | – | 2,867 (81.5) | 2,581 |
| SWs | 553 (45.6) | 1,268 (36.1) | Apartment | 1,101 (86.8) | 2,079 (80.6) |
| Taxi/truck drivers | 221 (18.2) | 551 (15.7) | Apartment | 442 (80.2) | 82 (3.2) |
| Pimps/hotspot managers | 110 (9.1) | 222 (6.3) | Apartment | 187 (84.2) | 98 (3.8) |
| Entertainment hotspot staff | 90 (7.4) | 202 (5.7) | Café/bar | 150 (74.3) | 40 (1.5) |
| NGO staff | 58 (4.8) | 727 (20.7) | Apartment | 587 (80.7) | 166 (6.4) |
| Law enforcement officers | 49 (4.0) | 137 (3.9) | Apartment | 116 (84.7) | 16 (0.6) |
| Massage parlor/sauna staff | 45 (3.7) | 101 (2.9) | Massage parlor/sauna | 72 (71.3) | 42 (1.6) |
| Hotel personnel | 38 (3.1) | 93 (2.6) | Hotel/motel | 65 (69.9) | 24 (0.9) |
| Other KIs (SW clients, data collectors) | 28 (2.3) | 109 (3.1) | Virtual hotspot | 68 (62.4) | 25 (1.0) |
| AIDS centers and social services staff | 20 (1.7) | 107 (3.0) | Street/park | 79 (73.8) | 8 (0.3) |

AIDS, acquired immunodeficiency syndrome; KI, key informant; NGO, non-governmental organization; SW, sex worker.

cross-type working patterns (Table 4). The five factors identified were: (1) Online & Independent Work, loading on virtual and escort hotspots; (2) Street-Apartment Axis, contrasting private apartments and street-based hotspots; (3) Hotel & Leisure Settings, combining hotels, saunas, and massage parlors; (4) Club-Mediated Work, including strip/art clubs, nightlife venues, and pimps; (5) Public-Facing Settings, defined by cafes and bars. Offices and highway hotspots did not align with any factor.

HIV service coverage was limited: 13.7% of hotspots were registered in SYREX, 34.1% received services, and 18.9% were reached by mobile vans. Office and street-based sites had the highest engagement; coverage was lowest in apartments, virtual platforms, and pimp-controlled venues.

Regression analysis, presented in Table 5, showed higher odds of HIV prevention service coverage in hotspots with 24/7 operation or non-balanced hours, open access, and the presence of SWs who inject drugs. For example, hotspots perceived as very safe had more than twice the odds of service coverage compared with neutral sites (aOR = 2.3; 95% CI: 1.3–3.9), and hotspots where SWs who inject drugs were present showed substantially higher coverage (aOR = 2.7; 95% CI: 1.6–4.4). Neither the presence of male SWs nor SEMs was significantly associated with coverage.

The estimated population of SW reported at hotspots in eight cities was 8,670 (range: 3,837–13,503), with 7,866 (90.7%) observed during 30-minute headcounts (Table 6). Counts varied by hotspot type and city (S1 Table). Apartments hosted the largest numbers (2,897 SWs), yet only 23.2% were estimated to be at registered hotspots, 48.3% at hotspots with reported service provision, and 28.0% at hotspots reached by mobile vans. Virtual venues (2,068 SWs) had the lowest overall coverage.

## Discussion

This study demonstrated that a multi-informant two-level mapping approach can effectively identify sex work hotspots and generate operationally relevant evidence for HIV prevention programming. Validation of 2,118 active hotspots revealed structural, temporal, and operational heterogeneity, with only about one-third receiving any prevention services. These findings highlight persistent access gaps and reinforce the importance of context-specific resource allocation, in line with UNAIDS' location-population approach [35] and spatial epidemiological frameworks [28].

**Table 3.  Characteristics of sex work hotspot types in eight Ukrainian cities, 2021.**

| Characteristics | Apart-ment | Virtual hotspot | Street/park | Escort/on-call | Massage parlor/sauna | Café/bar | Hotel/motel | Office ('brothel') | Night-club/casino | Route/high-way | Pimps | Art club/strip club | Over-all |
|---|---|---|---|---|---|---|---|---|---|---|---|---|---|
| | % | % | % | % | % | % | % | % | % | % | % | % | % |
| Reported hotspots, N | 1065 | 296 | 289 | 143 | 172 | 184 | 153 | 98 | 54 | 72 | 29 | 25 | 2580 |
| Validated active, % | 85.9 | 83.4 | 81.7 | 93.7 | 72.7 | 65.8 | 74.5 | 79.6 | 75.9 | 84.7 | 100.0 | 68.0 | 82.1 |
| **Total active, N** | **915** | **247** | **236** | **134** | **125** | **121** | **114** | **78** | **41** | **61** | **29** | **17** | **2118** |
| **Subgroup presence** | | | | | | | | | | | | | |
| SWs who inject drugs | 5.9 | 2.0 | 20.3 | 0.7 | 6.4 | 9.9 | 5.3 | 11.5 | 14.6 | 36.1 | – | 23.5 | 8.3 |
| Male SWs | 3.3 | 9.3 | 4.2 | 1.5 | 4.0 | 0.8 | 2.6 | 1.3 | 4.9 | – | 62.1 | 11.8 | 4.6 |
| SEMs | 2.4 | 4.5 | 10.6 | – | 10.4 | 20.7 | – | 2.6 | 19.5 | 8.2 | 3.4 | 11.8 | 6.9 |
| **Age of SWs** | | | | | | | | | | | | | |
| Younger (max age ≤ 25 years) | 10.6 | 13.4 | 5.9 | 45.5 | 8.0 | 19.0 | 12.3 | 1.3 | 14.6 | 1.6 | 6.9 | 41.2 | 12.7 |
| Mixed (varied age ranges) | 50.7 | 76.1 | 69.9 | 9.7 | 82.4 | 71.9 | 78.9 | 87.2 | 85.4 | 59.0 | 82.8 | 47.1 | 60.5 |
| Older (min age > 25 years) | 38.7 | 10.5 | 24.2 | 44.8 | 9.6 | 9.1 | 8.8 | 11.5 | – | 39.3 | 10.3 | 11.8 | 26.8 |
| **# SWs during L2-visit** | | | | | | | | | | | | | |
| ≤ 3 SWs | 59.9 | 53.8 | 42.8 | 95.5 | 50.4 | 64.5 | 46.5 | 24.4 | 22.0 | 60.7 | 41.1 | 23.5 | 55.9 |
| 4–9 SWs | 37.3 | 37.7 | 55.1 | 4.5 | 48.8 | 34.7 | 49.1 | 67.9 | 61.0 | 36.1 | 24.1 | 70.6 | 40.0 |
| ≥ 10 | 2.8 | 8.5 | 2.1 | – | 0.8 | 0.8 | 4.4 | 7.7 | 17.1 | 3.3 | 34.5 | 5.9 | 4.0 |
| **Season** | | | | | | | | | | | | | |
| Summer | 99.8 | 100.0 | 99.2 | 100.0 | 99.2 | 100.0 | 100.0 | 100.0 | 97.6 | 100.0 | 100.0 | 100.0 | 99.7 |
| Spring | 99.9 | 100.0 | 96.6 | 100.0 | 100.0 | 99.2 | 100.0 | 97.4 | 92.7 | 100.0 | 100.0 | 100.0 | 99.3 |
| Autumn | 99.6 | 100.0 | 86.4 | 100.0 | 100.0 | 96.7 | 98.2 | 97.4 | 97.6 | 95.1 | 100.0 | 100.0 | 97.7 |
| Winter | 99.1 | 99.6 | 71.2 | 100.0 | 97.6 | 92.6 | 97.4 | 97.4 | 82.9 | 85.2 | 100.0 | 100.0 | 94.8 |
| **Operate year-round** | **98.9** | **99.6** | **69.9** | **100.0** | **96.8** | **92.6** | **97.4** | **97.4** | **78.0** | **85.2** | **100.0** | **100.0** | **94.4** |
| **Working days** | | | | | | | | | | | | | |
| Monday | 96.2 | 94.7 | 88.6 | 100.0 | 97.6 | 89.3 | 98.2 | 100.0 | 68.3 | 85.2 | 55.2 | 76.5 | 93.8 |
| Tuesday | 98.7 | 96.0 | 98.3 | 100.0 | 100.0 | 97.5 | 100.0 | 100.0 | 85.4 | 100.0 | 55.2 | 94.1 | 97.7 |
| Wednesday | 99.1 | 96.8 | 98.7 | 100.0 | 100.0 | 98.3 | 99.1 | 100.0 | 90.2 | 100.0 | 55.2 | 100.0 | 98.2 |
| Thursday | 99.3 | 97.6 | 99.2 | 100.0 | 100.0 | 98.3 | 99.1 | 100.0 | 90.2 | 100.0 | 55.2 | 100.0 | 98.4 |
| Friday | 99.9 | 98.4 | 100.0 | 100.0 | 100.0 | 100.0 | 100.0 | 100.0 | 97.6 | 100.0 | 69.0 | 100.0 | 99.3 |
| Saturday | 98.4 | 97.6 | 100.0 | 100.0 | 100.0 | 97.5 | 100.0 | 100.0 | 97.6 | 100.0 | 100.0 | 100.0 | 98.8 |
| Sunday | 97.4 | 97.2 | 100.0 | 100.0 | 100.0 | 97.5 | 100.0 | 100.0 | 100.0 | 100.0 | 58.6 | 100.0 | 97.8 |
| **24/7 without days off** | **67.1** | **77.7** | **10.2** | **94.8** | **82.4** | **7.4** | **73.7** | **85.9** | **24.4** | **19.7** | **44.8** | **29.4** | **59.5** |
| **Schedule pattern** | | | | | | | | | | | | | |
| Predominantly nighttime | 20.3 | 14.2 | 74.6 | 5.2 | 9.6 | 43.8 | 22.8 | 7.7 | 70.7 | 55.7 | 44.8 | 70.6 | 27.8 |
| Predominantly daytime | 8.2 | 4.0 | 10.2 | – | 6.4 | 44.6 | 2.6 | 6.4 | 2.4 | 24.6 | – | – | 9.2 |
| Balanced | 71.5 | 81.8 | 15.3 | 94.8 | 84.0 | 11.6 | 74.6 | 85.9 | 26.8 | 19.7 | 55.2 | 29.4 | 63.0 |
| **Median work hours (IQR)** | | | | | | | | | | | | | |
| Weekday | 24 (12-24) | 24 (24–24) | 7 (5-10) | 24 (24–24) | 24 (24–24) | 11 (6-13) | 24 (12-24) | 24 (24–24) | 8 (6-18) | 7 (5-13) | 14 (0-24) | 9 (6-24) | 24 (10-24) |

*(Continued)*

**Table 3.**  (Continued)

| Characteristics | Apart-ment | Virtual hotspot | Street/park | Escort/on-call | Massage parlor/sauna | Café/bar | Hotel/motel | Office ('brothel') | Night-club/casino | Route/high-way | Pimps | Art club/strip club | Over-all |
|---|---|---|---|---|---|---|---|---|---|---|---|---|---|
| | % | % | % | % | % | % | % | % | % | % | % | % | % |
| Weekend | 24 (13-24) | 24 (24−24) | 7 (5-10) | 24 (24−24) | 24 (24−24) | 11 (6-13) | 24 (12-24) | 24 (24−24) | 8 (7-19) | 7 (6-12) | 24 (4-24) | 9 (7-24) | 24 (10-24) |
| **Open access** | **87.3** | **83.4** | **94.5** | **88.1** | **94.4** | **97.5** | **86.0** | **100.0** | **90.2** | **82.0** | **31.0** | **88.2** | **88.2** |
| **Safety level** | | | | | | | | | | | | | |
| Very or fairly safe | 86.2 | 84.6 | 67.4 | 98.5 | 47.2 | 35.5 | 62.3 | 80.8 | 61.0 | 37.7 | 96.6 | 52.9 | 76.0 |
| 50/50 | 10.7 | 13.4 | 16.1 | 1.5 | 44.8 | 43.0 | 32.5 | 16.7 | 31.7 | 27.9 | – | 35.3 | 17.2 |
| Very or fairly unsafe | 3.1 | 2.0 | 16.5 | – | 8.0 | 21.5 | 5.3 | 2.6 | 7.3 | 34.4 | 3.4 | 11.8 | 6.8 |
| **Cross-type hotspots** | **45.2** | **43.3** | **29.7** | **90.3** | **51.2** | **38.0** | **60.5** | **55.1** | **36.6** | **34.4** | **93.1** | **41.2** | **47.4** |
| **Registered with HIV program**[a] | **17.9** | **0.4** | **24.6** | **0.7** | **3.2** | **3.3** | **3.5** | **50.0** | **4.9** | **19.7** | **–** | **5.9** | **13.7** |
| **Reported HIV pre-vention coverage**[b] | **38.0** | **35.2** | **56.8** | **0.7** | **8.0** | **10.7** | **14.9** | **73.1** | **24.4** | **65.6** | **6.9** | **23.5** | **34.1** |
| **Reported HIV mobile van coverage**[b] | **19.5** | **4.9** | **46.2** | **–** | **4.8** | **8.3** | **10.7** | **47.4** | **17.1** | **44.3** | **–** | **11.8** | **18.9** |
| **HIV prevention category** | | | | | | | | | | | | | |
| Reported & registered | 16.9 | 0.4 | 21.6 | 0.7 | 3.2 | 3.3 | 3.5 | 50.0 | 4.9 | 18.0 | – | 5.9 | 12.9 |
| Reported & not registered | 21.1 | 34.8 | 35.2 | – | 4.8 | 7.4 | 11.4 | 23.1 | 19.5 | 47.5 | 6.9 | 17.6 | 21.2 |
| Not reported & registered | 1.0 | – | 3.0 | – | – | – | – | – | – | 1.6 | – | – | 0.8 |
| Not reported & not registered | 61.0 | 64.8 | 40.3 | 99.3 | 92.0 | 89.3 | 85.1 | 26.9 | 75.6 | 32.8 | 93.1 | 76.5 | 65.1 |

[a]'Registered with HIV program' indicates hotspots officially registered in SYREX as recipients of HIV prevention services in 2020.

[b]'Reported HIV prevention coverage' and 'Reported HIV mobile van coverage' refer to hotspots where key informants reported any form of HIV prevention service during the month prior to the visit.

IQR, interquartile range; L2, Level 2; SEM, sexually exploited minor; SW, sex worker.

Nearly half of identified hotspots involved SWs operating exclusively within a single venue type, underscoring a high degree of operational segmentation in Ukraine's sex work landscape. Similar patterns have been documented in earlier Ukrainian studies [39], in contrast to settings such as India or Mauritius, where SWs frequently alternate between streets, apartments, and other locations [57,58]. At the same time, cross-type organization was also observed, and factor analysis helped summarize these overlaps into a limited number of shared operational settings, rather than treating each venue category as fully distinct. From a programmatic perspective, these overlaps indicate that HIV prevention strategies organized around single venue categories may fail to capture entire operational networks of sex work. Instead, the findings point to the value of intervention models aligned with broader access environments that share common mechanisms of control, visibility, and risks [59]. For example, the "Hotel & Leisure Settings" factor – linking hotels/motels with massage parlors and saunas – captures a cluster of semi-formal indoor venues characterized by mediated entry and management control, suggesting the value of coordinated venue-level engagement. Such approaches may include negotiated access with venue owners and managers, on-site or proximate delivery of prevention services, and environmental risk reduction strategies that normalize condom use and support safer working conditions [59–62].

**Table 4. Varimax-rotated factor loadings for sex work hotspot types, 2021.**

| Hotspot Type | Factor 1. Online & Independent Work | Factor 2. Street-Apartment Axis | Factor 3. Hotel & Leisure Settings | Factor 4. Club-Mediated Work | Factor 5. Public-Facing Settings |
|---|---|---|---|---|---|
| Virtual hotspot | **0.717** | 0.090 | −0.082 | 0.016 | −0.051 |
| Escort/on-call | **0.536** | 0.175 | 0.046 | 0.233 | −0.136 |
| Apartment | −0.232 | **0.790** | −0.129 | 0.040 | −0.350 |
| Street/park | −0.203 | **−0.449** | −0.082 | 0.024 | −0.029 |
| Hotel/motel | −0.006 | 0.073 | **0.659** | 0.164 | −0.014 |
| Massage parlor/sauna | −0.086 | 0.036 | **0.558** | 0.132 | 0.050 |
| Art club/strip club | 0.039 | 0.019 | 0.097 | **0.458** | 0.058 |
| Nightclub/casino | −0.018 | −0.013 | 0.130 | *0.337* | 0.161 |
| Pimps | 0.181 | 0.037 | −0.063 | *0.324* | −0.098 |
| Café/bar | −0.115 | −0.003 | 0.041 | 0.086 | **0.544** |
| Office ('brothel') | 0.025 | −0.023 | 0.323 | −0.034 | 0.032 |
| Route/highway | −0.110 | −0.235 | −0.021 | −0.032 | −0.064 |

Extraction method: Principal Axis Factoring. Rotation method: Varimax with Kaiser Normalization. Rotation converged in 8 iterations. Loadings ≥0.40 are shown in bold.

Beyond venue clustering, the findings also point to a substantial presence of a highly privatized, decentralized, and low-visibility sex work environment. Historically visible settings – such as street-based sites and brothels – received the majority of programmatic attention, reflecting the long-standing focus of harm reduction efforts on SWs who inject drugs [63,64]. However, several dominant but less visible hotspot settings consistently exhibited markedly lower likelihood of HIV prevention service coverage. In particular, private apartments and Online & Independent Work settings showed the lowest coverage, indicating a systemic and predictable service gap rather than incidental under-reach. This aligns with prior evidence that autonomy, confidentiality, and dispersed organization in such settings limit the feasibility of traditional outreach approaches [63,65], even in contexts of elevated exposure to sexual violence [62]. Similar challenges have been reported across diverse settings, including sub-Saharan Africa, where programmatic mapping exercises have highlighted the limited reach of venue-based models among SWs operating in home-based or digitally mediated contexts [34]. While the global shift toward digitally mediated sex work is well documented across multiple settings and low- and middle-income countries [66–68], empirical evidence on the scale of Online & Independent Work settings and their exclusion from HIV services remains limited [37,68]. To our knowledge, this study is the first in Ukraine to systematically identify and classify digital venues within a broader hotspot mapping framework. Given the consistently low coverage observed in Online & Independent Work settings, these findings suggest that digital strategies should be considered a primary – rather than supplementary – modality of HIV prevention for this segment. However, evidence-based approaches such as virtual outreach through online platforms, HIV self-testing distribution through digital tools, and digital navigation to testing and treatment services [69–71] remain largely absent from Ukraine's programming [72], with only limited small-scale interventions [11,73,74]. The expansion of digital sex work settings – accelerated by the 2022 full-scale Russian invasion, venue restrictions, and mobility constraints under curfew [16] – is unlikely to reverse, underscoring the urgency of systematic digital outreach to prevent widening service gaps. Taken together, these findings point to a structural policy gap: national HIV programs need to formally recognize digital outreach as a core prevention modality, supported by dedicated funding, performance indicators, and clear guidance on confidentiality, digital safety, and linkage to care.

HIV prevention service coverage also varied systematically across other hotspot characteristics, reflecting broader structural and contextual barriers. Indoor or formalized hotspots – including Hotel & Leisure Settings and Club-Mediated Work environments – were less frequently covered, reflecting the combined effects of managerial control, mediated

**Table 5. Associations of hotspot characteristics with HIV prevention service coverage.**

| Predictor | aOR | 95% CI | | p-value |
|---|---|---|---|---|
| | | Lower | Upper | |
| **Hotspot type** | | | | |
| Street/park (Reference) | 1.0 | – | – | – |
| Office ('brothel') | 2.3 | 1.0 | 5.2 | 0.052 |
| Route/highway | 3.6 | 1.4 | 9.4 | **0.010** |
| Café/bar | 0.2 | 0.1 | 0.4 | **<0.001** |
| Apartment | 0.2 | 0.1 | 0.4 | **<0.001** |
| Art club + nightclub + pimps (Club-Mediated Work) | 0.2 | 0.1 | 0.4 | **<0.001** |
| Hotel + massage parlor/sauna (Hotel & Leisure Settings) | 0.2 | 0.1 | 0.3 | **<0.001** |
| Virtual + escort/on-call (Online & Independent Work) | 0.1 | 0.1 | 0.3 | **<0.001** |
| **24/7 without days off (yes vs. no)** | 2.2 | 1.1 | 4.2 | **0.025** |
| **Schedule type** | | | | |
| Balanced (Reference) | 1.0 | – | – | – |
| Predominantly nighttime | 2.2 | 1.1 | 4.2 | **0.021** |
| Predominantly daytime | 2.8 | 1.4 | 5.9 | **0.005** |
| **# SWs during L2-visit** | | | | |
| 4–9 SWs (Reference) | 1.0 | – | – | – |
| ≤3 SWs | 0.7 | 0.5 | 0.9 | **0.014** |
| ≥10 SWs | 0.4 | 0.2 | 0.9 | **0.023** |
| **Safety level** | | | | |
| 50/50 (Reference) | 1.0 | – | – | – |
| Very unsafe | 1.1 | 0.2 | 5.5 | 0.928 |
| Fairly unsafe | 0.2 | 0.1 | 0.5 | **<0.001** |
| Fairly safe | 1.1 | 0.7 | 1.7 | 0.694 |
| Very safe | 2.3 | 1.3 | 3.9 | **0.004** |
| **Open access (yes vs. no)** | 2.1 | 1.1 | 4.0 | **0.018** |
| **Presence of SWs who inject drugs (yes vs. no)** | 2.7 | 1.6 | 4.4 | **<0.001** |
| **Presence of male SWs (yes vs. no)** | 0.7 | 0.3 | 1.3 | 0.234 |
| **Presence of SEMs (yes vs. no)** | 0.5 | 0.3 | 1.1 | 0.071 |

Multivariable generalized linear mixed model with binomial distribution and logit link. Random intercept for city to account for clustering. Significant associations (p < 0.05) are shown in bold. Reference categories are shown with an odds ratio of 1.0.

aOR, adjusted odds ratio; CI, confidence interval; SEM, sexually exploited minor; SW, sex worker.

access, and institutional gatekeeping, a pattern reported in both global [62] and Ukrainian contexts [64]. These access constraints limited outreach to closed venues and contributed to persistent service gaps. In contrast, hotspots where SWs who inject drugs were present were more likely to receive prevention services, indicating that long-standing programmatic prioritization of this subgroup continues to shape service reach across settings. While such individuals were more common at public and highway venues, their presence within manager-controlled environments highlights overlooked prevention needs among less visible segments of this high-risk population. This challenges the assumption – derived from studies in high-income settings such as the UK – that SWs who inject drugs are concentrated exclusively in street-based settings [61], and points to a visibility bias in outreach planning.

Perceived safety emerged as another key determinant of service access. Hotspots described as 'very safe' were twice as likely to be reached by prevention teams, underscoring the role of trust, regulation, and perceived risk in enabling outreach [75]. Conversely, cafés and bars (Public-Facing Settings) showed substantially lower coverage despite their public

**Table 6. Estimated number of SWs at hotspots in eight Ukrainian cities, 2021.**

| Hotspot Type | Estimated number of SWs | | | | | | | | # SWs observed during L2-visit |
| | Total | | Registered with HIV program[a] | | Reported HIV prevention coverage[b] | | Reported HIV mobile van coverage[b] | | |
| | Midpoint | Min-Max | Midpoint (%)³ | Min-Max | Midpoint (%)³ | Min-Max | Midpoint (%)³ | Min-Max | N (% observed)[c] |
|---|---|---|---|---|---|---|---|---|---|
| **Total, N** | **8,670** | **3,837−13,503** | **1,303 (15.0)** | **703−1,903** | **3,161 (36.5)** | **1,492−4,710** | **1,930 (22.3)** | **916−2,944** | **7,866 (90.7)** |
| Apartment | 2,897 | 1,540−4,254 | 673 (23.2) | 382-964 | 1,399 (48.3) | 719−2,078 | 811 (28.0) | 391−1,230 | 3,164 (109.2) |
| Virtual hotspot | 2,068 | 520−3,615 | 6 (0.3) | 2-10 | 307 (14.8) | 97-517 | 93 (4.5) | 22-163 | 1,088 (52.6) |
| Street/park | 919 | 446-1392 | 271 (29.5) | 158-384 | 561 (61.0) | 280-841 | 470 (51.1) | 233-706 | 951 (103.5) |
| Office ('brothel') | 459 | 207-711 | 229 (49.8) | 98-359 | 354 (77.1) | 153-554 | 242 (52.6) | 102-381 | 409 (89.1) |
| Massage parlor/ sauna | 446 | 199-692 | 18 (4.0) | 8-28 | 43 (9.6) | 24-61 | 29 (6.5) | 17-41 | 463 (103.9) |
| Hotel/motel | 434 | 201-666 | 22 (5.1) | 13-31 | 85 (19.6) | 48-121 | 63 (14.5) | 35-91 | 443 (102.2) |
| Pimps | 364 | 134-594 | – | – | 49 (13.5) | 18-80 | – | – | 265 (72.8) |
| Café/bar | 357 | 180-533 | 13 (3.5) | 6-19 | 49 (13.7) | 22-75 | 37 (10.4) | 17-57 | 380 (106.6) |
| Nightclub/casino | 251 | 120-382 | 13 (5.2) | 7-19 | 102 (40.6) | 57-147 | 89 (35.3) | 50-127 | 230 (91.6) |
| Route/highway | 207 | 95-319 | 50 (23.9) | 22-77 | 133 (64.3) | 60-206 | 90 (43.5) | 45-135 | 217 (104.8) |
| Escort/on-call | 173 | 144-201 | 3 (1.7) | 2-4 | 3 (1.7) | 2-4 | – | – | 171 (99.1) |
| Art club/strip club | 98 | 51-144 | 7 (6.7) | 5-8 | 19 (19.4) | 12-26 | 9 (8.7) | 4-13 | 85 (87.2) |

[a]'Registered with HIV program' indicates hotspots officially registered as recipients of HIV prevention services in 2020.

[b]'Reported HIV prevention coverage' and 'Reported HIV mobile van coverage' refer to hotspots where key informants reported any form of HIV prevention service during the month prior to the visit.

[c]Percentages represent the proportion of total estimated number in the first numeric column.

HIV, human immunodeficiency virus; L2, Level 2; SW, sex worker.

visibility, suggesting that visibility alone does not translate into service access. These venues were frequently described as unsafe environments, where alcohol use is common and condomless sex commands higher prices [62], further constraining outreach feasibility. Importantly, prevention coverage did not correlate with hotspot size: a substantial proportion of Club-Mediated Work settings hosted over 10 SWs, yet these higher-volume locations were less likely to receive services. Together, these findings suggest that structural and gatekeeping barriers may outweigh epidemiological considerations such as venue size or visibility in shaping prevention coverage.

Working hours further influenced prevention service access. Hotspots open 24/7 or primarily during daytime hours were more likely to be reached than those with 'balanced' schedules, likely reflecting NGO resource constraints and limited capacity for night-time outreach. Security risks after dark – well documented in other settings including the Democratic Republic of Congo [76] – are particularly acute in Ukraine, where martial law, curfews, and transport limitations restrict safe mobility. Additionally, 24.1–34.2% of cafes, massage parlors, hotels, and nightclubs were inactive at the time of mapping, likely reflecting long-term disruptions to sex work infrastructure because of the COVID-19 pandemic. Comparable COVID-19-related disruptions have been reported across Central and Eastern Europe and Central Asia, where lockdowns and venue closures reshaped sex work organization, reduced safety, and limited outreach feasibility across both indoor and outdoor settings [77]. Taken together, these findings suggest that temporal misalignment between outreach operations and actual hotspot activity constitutes a structural barrier to service coverage. Addressing this gap requires outreach models that are responsive to real operating hours, including adequately resourced evening outreach or flexible-hour models, implemented with appropriate safety measures and coordination with municipal services. In this context, network-based strategies – already implemented in Ukraine [78,79] – have demonstrated promising results in reaching SWs who inject drugs and those operating in concealed or manager-controlled venues

[80]. These community-led models adapt to real working conditions rather than rely on fixed schedules or venue-based access. By leveraging peer trust and existing social networks, such approaches extend coverage beyond conventional outreach and help mitigate structural access barriers. A combined model integrating peer recruitment with targeted, time-responsive outreach therefore represents a pragmatic pathway to closing service gaps across diverse sex work settings.

The identified structural challenges also reinforce the critical importance of trust-based partnerships with SW communities. The multi-informant mapping approach, grounded in active participation of SWs as interviewers, KIs, and gatekeepers, achieved a high validation rate (86.8%), with hotspots reported by SWs showing higher confirmation compared to those identified by other informant groups, demonstrating the reliability of community-generated data and the added value of participatory methods. Community engagement was especially effective in identifying hidden or stigmatized venues – a pattern previously documented in community-based screening initiatives in the United States [19] – which were less frequently captured by institutional informants, such as AIDS center staff. Without SW involvement, mapping efforts risk misrepresenting the operational landscape and missing populations most in need of services. Scaling community-led outreach is therefore not only methodologically sound but programmatically strategic. In this study, community participation directly improved data completeness and accuracy, particularly for less visible and privatized settings. Modeling evidence from Ukraine suggests that increasing the share of services delivered by community-based organizations to 65% could avert approximately 2,220 HIV infections among SWs – a 12% reduction [81], aligning with global 2025 targets that emphasize community leadership in HIV prevention and mapping [82]. Integrated models combining hotspot mapping, mobile testing, and community mobilization have shown effectiveness across multiple settings, including Peru [17], sub-Saharan Africa and Thailand [20], and other outreach contexts [76], and could greatly improve reach in Ukraine's criminalized and under-resourced environments.

Finally, the study revealed substantial underreporting of HIV prevention service coverage within Ukraine's national monitoring SYREX registry. Only a small fraction (13.7%) of validated hotspots were recorded in SYREX, while KIs reported service delivery at a considerably larger share of locations (34.1%). This discrepancy reflects not only incomplete reporting but also systemic limitations in the data collection system – most notably, the absence of mandatory geolocation. A major contributor is off-site outreach: as in other countries (e.g., in Uganda [25]), Ukrainian teams often meet SWs from indoor or virtual venues at neutral locations (e.g., streets, NGO offices) to protect privacy and minimize legal risk. However, these interactions are rarely linked back to the original hotspot, leaving many serviced venues effectively invisible in official records. Introducing an 'alternative outreach location' field with geospatial tagging would improve monitoring accuracy and data completeness. Additional underreporting arises when services are arranged by managers or intermediaries, particularly within structured environments such as the Club-Mediated Work settings [24,39]. In such cases, services may occur outside NGO oversight and remain unregistered. Additionally, time lags between fieldwork (2021) and registry data (2020) may also contribute to this. However, the fact that even highly visible public hotspots – such as highways – were frequently missing in the registry suggests systemic reporting failures. These findings point to the need to revise routine monitoring systems – particularly by introducing mandatory geolocation fields, standardizing reporting for off-site outreach, and strengthening accountability for data completeness – so that resource allocation reflects on-the-ground service delivery.

## Limitations

Despite mapping 2,118 hotspots and reaching saturation, our findings likely underestimate the full scale and complexity of the sex work landscape. Even without accounting for mobility [49], discrepancies between hotspot-based population estimates generated in this study and prior population estimates (Table 1) reflect well-documented limitations of hotspot-based methods in capturing hidden, fluid, or stigmatized subgroups of people involved in sex work [83]. Several structural and contextual factors contribute to this underestimation.

One key factor is the exclusion of transactional sex – defined as informal exchanges without clear pricing or self-identification as a SW [84]. These exchanges often occur outside formal venues yet may spatially overlap with mapped hotspots [49]. However, the mapping protocol required confirmation that observed activities met the operational definition of sex work, excluding informal transactional exchanges that did not involve self-identification as sex work. Economic instability and war-related displacement may further increase reliance on such informal exchanges, particularly among structurally vulnerable women [15,85].

Another challenge involves the limited visibility of online-mediated sex work. While the study aimed to capture digital venues, current tools remain insufficient for mapping such activity [30]. Many SWs operate independently or outside established networks [58], often avoiding the 'commercial' label to minimize stigma or legal risk [57]. This invisibility is especially pronounced among male SWs – estimated to comprise up to 10% of Ukraine's SW population [7] – who are more likely to use digital platforms due to layered stigmas, including those related to same-sex behavior [86,87]. From a programmatic perspective, this limitation suggests that physical mapping should be complemented with digital outreach and monitoring strategies that better capture independent and platform-mediated sex work, while addressing privacy, stigma, and legal risk.

Most critically, the criminalization of sex work continues to drive invisibility. Legal risks and punitive policing compel SWs to conceal their activities or relocate frequently – dynamics consistently documented across diverse settings, including sub-Saharan Africa [34,49]. As a result, many hotspots remained unobserved and underserved [24,62,81]. From a policy perspective, these dynamics underscore how punitive legal environments undermine both surveillance and service delivery [88], suggesting that engagement with law enforcement and municipal authorities around harm reduction-oriented practices is essential to improving both data completeness and access to HIV prevention services. Our findings should therefore be interpreted as conservative lower-bound estimates. While the current mapping offers the most comprehensive pre-war snapshot to date, future strategies must better capture digital, informal, and transactional forms of sex work to reach underserved subpopulations and improve HIV responses.

This study represents only a snapshot and does not account for temporal fluctuations. For example, street-based activity declined in winter, but it remains unclear whether SWs paused work or transitioned online – raising the possibility that certain venue types were underrepresented. In addition, hotspot schedules were classified broadly, limiting insights into peak hours or seasonal variations [89] – factors critical for effective outreach planning. These limitations point to the importance for implementing partners and funders of supporting flexible, community-informed outreach models that can adapt to temporal shifts in sex work patterns, rather than relying on static assumptions about venue type or operating hours.

Although this represents the most extensive hotspot mapping conducted in Ukraine to date, it predates the 2022 full-scale Russian invasion. Since then, mass displacement [16,90], venue closures, curfews [15], economic hardship [91], and the rapid shift to online solicitation [15] have fundamentally reshaped sex work environments. Evidence suggests that SWs may now concentrate in locations with a high density of single men [30], especially as women with children leave the country and military personnel become a prevalent client group [91,92]. These structural shifts may alter both risk exposure and service delivery feasibility in ways not captured by our data.

Despite these constraints, the study offers a methodological framework for systematic, community-informed hotspot mapping and establishes a robust pre-war baseline that can serve as a critical benchmark for future comparative analyses. The utility of future mapping efforts could be strengthened by collecting additional data on structural determinants of service access and vulnerability, including interactions with venue management, access constraints, and SWs' perspectives on privacy, fear of exposure, or police presence [25,87]. Data on HIV risk levels across hotspot types [61], and working conditions such as autonomy, coercion, and protection mechanisms [62,89], would enable more nuanced analyses.

To enhance responsiveness in a shifting context, incorporating hotspot mapping into routine HIV program monitoring could enable real-time detection of changes in hotspot typologies, geographic clustering, and service reach [49]. While

full-scale mapping is resource-intensive [83] and limited by funding [93,94], periodic micro-mapping targeting dynamic areas or underserved populations offers a feasible alternative. Overlaying hotspot maps with service data can help identify 'service deserts' [95] and guide strategic deployment of mobile units, peer navigations, or digital tools. Geospatial modeling based on environmental features – rather than static addresses – may further improve adaptability in contexts of displacement and war, as shown in other settings [96]. For national health authorities and donors, these approaches offer a pragmatic means to strengthen routine monitoring, detect emerging service gaps earlier, and support more adaptive and context-sensitive allocation of outreach resources.

## Conclusion

This study presents the most comprehensive mapping of sex work hotspots in Ukraine to date, revealing numerous previously undocumented venues, diverse operational typologies, and substantial gaps in HIV prevention service coverage. In particular, the low coverage in private apartments, virtual platforms, and other low-visibility settings highlights structural blind spots in current outreach models and underscores the need for digital and network-based prevention strategies. The high validation rate achieved through community engagement demonstrates the value of community-led mapping for identifying hidden and rapidly evolving sex work environments. Evidence from this study suggests that integrating routine hotspot mapping into HIV program monitoring – alongside strengthened community-led outreach and digital service delivery – could substantially improve the alignment between service provision and on-the-ground-realities. As Ukraine's HIV response continues under rapidly shifting war-related and epidemiological conditions, hotspot mapping should become a core component of adaptive, data-driven HIV programming. Linking mapped hotspot data with service delivery records, including for off-site and digital outreach, is essential to improving resource allocation, reducing service gaps, and accelerating progress toward national and global HIV goals.

## Supporting information

**S1 File. Data collection tool – hotspot passport.**
(PDF)

**S1 Table. Characteristics of sex work hotspots in eight Ukrainian cities, 2021.**
(PDF)

**S2 Table. Heatmap of hourly activity by hotspot type in eight Ukrainian cities, 2021.**
(PDF)

## Acknowledgments

The authors sincerely thank the SW community members and hotspot staff for their openness, trust, and invaluable contribution to this study. We are also grateful to the key informants and gatekeepers whose cooperation was essential to the successful mapping process. Special thanks go to the regional mapping teams led by Serhii Stratulat, Yulia Tsarevska, Olena Zhuk, Lidiia Zvarych, Olena Duz, Svitlana Sirenko, Natalia Chantseva, and Petro Syvokin, as well as to the NGO specialists involved, for their dedication and proactive support throughout the research. We also deeply appreciate the committed team from Alliance Consultancy, whose expertise and efforts were instrumental in conducting the mapping and data collection.

## Author contributions

**Conceptualization:** Oksana Kovtun.

**Data curation:** Oksana Kovtun, Olga Cheshun.

**Formal analysis:** Oksana Kovtun, Kostyantyn Dumchev.

**Investigation:** Oksana Kovtun.

**Methodology:** Oksana Kovtun.

**Project administration:** Oksana Kovtun.

**Software:** Olga Cheshun.

**Validation:** Olga Cheshun, Oksana Pashchuk.

**Writing – original draft:** Oksana Kovtun.

**Writing – review & editing:** Oksana Kovtun, Olga Cheshun, Oksana Pashchuk, Kostyantyn Dumchev.

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
