## [Decision Letter · Decision Letter 0]

22 Dec 2025

Dear Dr. Kovtun,

Thank you for submitting your manuscript to PLOS ONE. After careful consideration, we feel that it has merit but does not fully meet PLOS ONE’s publication criteria as it currently stands. Therefore, we invite you to submit a revised version of the manuscript that addresses the points raised during the review process.

**ACADEMIC EDITOR:**plosone@plos.org . Also review your citations, figures and tables and ensure they fully comply with plosone requirements.

We look forward to receiving your revised manuscript.

Kind regards,

Ibrahim Jahun, MD, MSC, PhD

Academic Editor

PLOS One

2. Thank you for stating the following in the Sources of support Section of your manuscript:

“This study was conducted with the financial support of the International Charitable Foundation “Alliance for Public Health” through the project “Improving HIV Treatment Cascade for Key Populations through Differentiated Case Detection and Linkage to Care and Increased Capacity at the Center for Public Health and Strategic Information in Ukraine”, funded by the U.S. Centers for Disease Control and Prevention (CDC) through the US President’s Emergency Plan for AIDS Relief (PEPFAR). This study is supported by the Cooperation Agreement NU2GGH002114 with the CDC. All responsibility for the content shall be borne by the authors, and the article does not necessarily reflect the official position of the CDC.”

“This study was conducted with the financial support of the International Charitable Foundation “Alliance for Public Health” through the project “Improving HIV Treatment Cascade for Key Populations through Differentiated Case Detection and Linkage to Care and Increased Capacity at the Center for Public Health and Strategic Information in Ukraine”, funded by the U.S. Centers for Disease Control and Prevention (CDC) through the US President’s Emergency Plan for AIDS Relief (PEPFAR). This study is supported by the Cooperation Agreement NU2GGH002114 with the CDC. All responsibility for the content shall be borne by the authors, and the article does not necessarily reflect the official position of the CDC.”

3. For studies involving third-party data, we encourage authors to share any data specific to their analyses that they can legally distribute. PLOS recognizes, however, that authors may be using third-party data they do not have the rights to share. When third-party data cannot be publicly shared, authors must provide all information necessary for interested researchers to apply to gain access to the data. (https://journals.plos.org/plosone/s/data-availability#loc-acceptable-data-access-restrictions)

Reviewers' comments:

Reviewer's Responses to Questions

**Comments to the Author**

1. Is the manuscript technically sound, and do the data support the conclusions?

Reviewer #1: Yes

Reviewer #2: Partly

2. Has the statistical analysis been performed appropriately and rigorously?

Reviewer #1: Yes

Reviewer #2: Yes

3. Have the authors made all data underlying the findings in their manuscript fully available?

Reviewer #1: No

Reviewer #2: Yes

4. Is the manuscript presented in an intelligible fashion and written in standard English?

Reviewer #1: Yes

Reviewer #2: Yes

Reviewer #1: • Generally, the topic addresses a critical gap in Ukraine's HIV response, where the epidemic is concentrated among key populations like SWs .

• The study incorporates modern challenges like digitalization (e.g., virtual platforms at 11.7%), which aligns with post-2020 shifts due to COVID-19 and conflict.

• The two-stage multi-informant design (1,212 secondary + 2,277 primary informants) is robust and community-led, achieving high validation (86.8%). Analysis includes advanced stats (e.g., multivariable mixed-effects regression), which is appropriate for clustered data like hotspots. The method is sound.

• The study will generally have a Policy Impact if the findings are utilized. Emphasizes equity for vulnerable subgroups, which will provide a better UNAIDS 95-95-95 and leave no one behind.

• However, there is a Lack of Novelty Clarity: While the study highlights "the most comprehensive mapping," it doesn't explicitly compare to prior work (e.g TAMPEP 2018 mapped FSW coverage at 40.3%). Without citing baselines, the contribution feels understated.

• The author needs to specify cities e.g Kyiv, Odesa?, sampling details (e.g., how informants were recruited), or ethical considerations e.g SW safety in conflict zones). Statistical methods are mentioned but not justified e.g why exploratory factor analysis?

• On the results, Percentages are precise, but no confidence intervals or p-values, reducing interpretability. Low coverage (e.g 13.7% registered) is highlighted, but no breakdown by city or subgroup size estimates.

• The paper Asserts "community-led mapping achieved high validation”, but doesn't quantify community involvement. Recommendations e.g strong engagement with SW communities, are a bit generic without linking to specific findings.

• There are Minor issues like inconsistent abbreviations (SWs vs. SW subgroups) and passive voice in places (e.g. "Data were analyzed" could be active for impact).

• Finally, no keywords or funding mention, common in abstracts.

• Recommendations for Improvement

• To strengthen this for submission, focus on clarity, depth, and enhance background: Add a sentence on Ukraine-specific context (e.g."Amid ongoing conflict, SW mobility has increased, complicating outreach"). Include a global benchmark (e.g."Global FSW HIV prevention coverage is approximately 50%, per UNAIDS").

• Refine Methods: Specify cities, informant recruitment (e.g., snowball sampling), and ethics (e.g., IRB approval). Justify analyses (e.g., "Mixed-effects regression accounted for city-level clustering").

• Add key stats with precision (e.g. say "Coverage OR=2.5 [95% CI:1.8-3.4] for safe hotspots").

• Strengthen Conclusions: Link directly to results (e.g., "Gaps in virtual hotspots (11.7% of total) underscore digital outreach needs"). Add implications (e.g., "Scaling community-led mapping could increase coverage by 20-30%, per similar studies").

• Overall, this is a solid paper with high public health relevance. With these tweaks, it could be publishable and impactful for HIV programs in conflict-affected areas.

Reviewer #2: General Comments

This manuscript, “Typologies of Sex Work Hotspots and Associations with HIV Prevention Coverage in Eight Ukrainian Cities: A Cross-Sectional Study,” makes a valuable contribution to public health by characterizing sex work hotspots and their relationship to HIV service coverage in Ukraine. The methodology is sound, and the data provides significant value to researchers, policymakers, and public health practitioners. The manuscript is well-written, and the following suggestions are offered to strengthen it further.

Major Revisions

Alignment of Objectives, Methods, and Findings: The manuscript would benefit from a clearer alignment between objectives, methods, and findings. The introduction of Exploratory Factor Analysis (EFA) in the results feels abrupt. To improve coherence:

a. Explicitly state research questions at the end of the Introduction, e.g., “1) What are the operational typologies of sex work hotspots in urban Ukraine? 2) What hotspot characteristics are associated with HIV prevention service coverage?”

b. In the Data Analysis section, justify each statistical method by linking it to a specific research question. For example, state that EFA was used to answer question 1 by identifying latent structures among hotspot types, and the Generalized Linear Mixed Model (GLMM) was used to answer question 2.

Rationale and Integration of the Factor Analysis: Strengthen the rationale for EFA and integrate its results more fully into the discussion.

a. In the Methods, justify the importance of uncovering these underlying factors and the research gap this analysis fills.

b. Enhance the Discussion section by exploring the programmatic implications of the factors beyond restating them. For example, discuss what the “Hotel & Leisure Settings” factor implies for intervention design, such as a unified outreach strategy targeting venue owners or highlighting a specific risk environment.

Strengthening the Discussion and Policy Implications: Strengthen the Discussion by drawing clearer connections between results and real-world implications.

a. Directly link specific findings from the regression analysis to actionable recommendations. For instance, for "Virtual + Escort/on-call" hotspots with extremely low coverage, recommend specific, evidence-based digital outreach strategies.

b. Increase international relevance by comparing identified hotspot typologies with those found in other settings (e.g., other parts of Eastern Europe or globally).

c. Sharpen policy messages with concrete recommendations for different stakeholders (e.g., NGOs, municipal health departments, law enforcement).

Minor Revisions

Clarity of the Regression Model in the Methods: In the Data Analysis section, clarify the hypothesis tested by the logistic regression model. Add a sentence stating that the model was used to identify which hotspot characteristics (independent variables) predict the likelihood of HIV prevention service coverage (the dependent variable).

Presentation of Reference Groups in Tables: In Table 5, include the reference categories for predictors (e.g., “street/park” for Hotspot type) as a line item in the table with an assigned Odds Ratio of “1.00 (Reference)”.

Self-Contained Tables: Ensure all abbreviations (e.g., aOR, CI, KMO) and technical terms within the tables are defined in the footnotes of each respective table.

**Do you want your identity to be public for this peer review?** For information about this choice, including consent withdrawal, please see our Privacy Policy

Reviewer #1: **Yes:** Helgar Musyoki

Reviewer #2: **Yes:** AliAkbar Haghdoost

---

## [Author Response · Author response to Decision Letter 1]

14 Jan 2026

Please see the uploaded document titled “Response to Reviewers” for a detailed, point-by-point response to all editor and reviewer comments.

Reviewer and editor comments are reproduced verbatim in that document, followed by our responses in italicized text. All changes to the manuscript are reflected in the revised version submitted.

---

## [Decision Letter · Decision Letter 1]

11 Feb 2026

Mapping of sex work hotspots to guide targeted HIV prevention: evidence from eight Ukrainian cities

PONE-D-25-51510R1

Dear Dr. Kovtun,

We’re pleased to inform you that your manuscript has been judged scientifically suitable for publication and will be formally accepted for publication once it meets all outstanding technical requirements. We haven't heard from 2nd reviewer, I have however reviewed your responses to their comments and judged them appropriate.

Kind regards,

Ibrahim Jahun, MD, MSC, PhD

Academic Editor

PLOS One

Additional Editor Comments (optional):

Reviewers' comments:

Reviewer's Responses to Questions

**Comments to the Author**

Reviewer #1: All comments have been addressed

2. Is the manuscript technically sound, and do the data support the conclusions?

Reviewer #1: Yes

3. Has the statistical analysis been performed appropriately and rigorously?

Reviewer #1: Yes

4. Have the authors made all data underlying the findings in their manuscript fully available?

Reviewer #1: Yes

5. Is the manuscript presented in an intelligible fashion and written in standard English?

Reviewer #1: Yes

Reviewer #1: I recommend acceptance for publication in PLOS ONE with no further major revisions. The authors have effectively addressed all previous comments, enhancing novelty, clarity, and policy relevance. This work has high public health impact, particularly for key population programming in Eastern Europe and conflict settings. If minor copyediting is needed (e.g., final consistency checks), it can be handled in production.

**Do you want your identity to be public for this peer review?** For information about this choice, including consent withdrawal, please see our Privacy Policy

Reviewer #1: **Yes:** Helgar Musyoki

---

## [Editor Report · Acceptance letter]

PONE-D-25-51510R1

PLOS One

Dear Dr. Kovtun,

I'm pleased to inform you that your manuscript has been deemed suitable for publication in PLOS One. Congratulations! Your manuscript is now being handed over to our production team.

Kind regards,

on behalf of

Dr. Ibrahim Jahun

Academic Editor

PLOS One